# Variability of cost trajectories over the last year of life in patients with advanced breast cancer in the Netherlands

Paul P. Schneider[1,2]*, Xavier G. L. V. Pouwels[1], Valéria Lima Passos[3], Bram L. T. Ramaekers[1], Sandra M. E. Geurts[4], Khava I. E. Ibragimova[4], Maaike de Boer[4], Frans Erdkamp[5], Birgit E. P. J. Vriens[6], Agnes J. van de Wouw[7], Marien O. den Boer[8], Manon J. Pepels[9], Vivianne C. G. Tjan-Heijnen[4], Manuela A. Joore[1]

1 Department of Clinical Epidemiology and Medical Technology Assessment (KEMTA), Maastricht University Medical Centre+, Maastricht, The Netherlands, 2 School of Health and Related Research, University of Sheffield, Sheffield, United Kingdom, 3 Department of Methodology and Statistics, CAPHRI Care and Public Health Research Institute Maastricht University, Maastricht, The Netherlands, 4 Department of Medical Oncology, GROW – School of Oncology and Developmental Biology, Maastricht University Medical Centre+, Maastricht, The Netherlands, 5 Zuyderland medical center, Sittard-Geleen, The Netherlands, 6 Catharina hospital, Eindhoven, The Netherlands, 7 VieCuri Medical Center, Venlo, The Netherlands, 8 Laurentius hospital Roermond, The Netherlands, 9 Elkerliek Hospital, Helmond, The Netherlands

* p.schneider@sheffield.ac.uk

**Data Availability Statement:** The data contains potentially identifiable patient information and cannot be shared publicly due to ethical and legal

## Abstract

### Objective

In breast cancer patients, treatment at the end of life accounts for a major share of medical spending. However, little is known about the variability of cost trajectories between patients. This study aims to identify underlying latent groups of advanced breast cancer patients with similar cost trajectories over the last year before death.

### Methods

Data from deceased advanced breast cancer patients, diagnosed between 2010 and 2017, were retrieved from the Southeast Netherlands Advanced Breast Cancer (SONABRE) Registry. Costs of hospital care over the last twelve months before death were analyzed, and the variability of longitudinal patterns between patients were explored using group-based trajectory modeling. Descriptive statistics and multinomial logistic regression were applied to investigate differences between the identified latent groups.

### Results

We included 558 patients. Over the last twelve months before death, mean hospital costs were €2,255 (SD = €492) per month. Costs increased over the last five months and reached a maximum of €3,614 in the last month of life, driven by hospital admissions, while spending for medication declined over the last three months of life. Based on patients' individual cost trajectories, we identified six latent groups with distinct longitudinal patterns, of which only two showed a marked increase in costs over the last twelve months before death. Latent

restrictions. However, the data of the SONABRE registry is organized according to the FAIR principles and researchers who meet the criteria for access to confidential data can submit their requests for data (and the R source code of this study) to the registry's PI prof. V.C.G. Tjan-Heijnen (vcg.tjan.heijnen@mumc.nl).

**Funding:** The data collection of the Southeast Netherlands Advanced Breast Cancer Registry was funded by the Netherlands Organization for Health Research and Development (ZonMw), Novartis, Pfizer, Roche Nederland B.V., Eisai B.V., and Eli Lilly NL. This work was supported by funding from Wellcome [108903/B/15/Z] and the University of Sheffield. The funders had no role in study design, data collection and analysis, decision to publish, or preparation of the manuscript.

**Competing interests:** The data collection of the Southeast Netherlands Advanced Breast Cancer Registry was funded by the Netherlands Organization for Health Research and Development (ZonMw), Novartis, Pfizer, Roche Nederland B.V., Eisai B.V., and Eli Lilly NL. The funders had no role in study design, data collection and analysis, decision to publish, or preparation of the manuscript, and this does also not alter our adherence to PLOS ONE policies on sharing data and materials.

groups were constituted of heterogeneous patients, and clinical characteristics explained membership only to a limited extent.

## Conclusions

The average costs of advanced breast cancer patients increased towards the end of life. However, we uncovered several latent groups of patients with divergent cost trajectories, which did not reflect the overall increasing trend. The mechanisms underlying the variability in cost trajectories warrants further research.

## Introduction

Advanced breast cancer (ABC) is a common type of cancer among women and a leading cause of cancer death worldwide [1,2]. Even though some patients live with ABC for many years, the disease is considered incurable, and the main objective of care is to prolong survival and sustain quality of life. Due to its high prevalence and high individual treatment costs, the economic burden of ABC in the Netherlands, as well as in many other countries is substantial [3,4].

A large share of the lifetime costs of ABC are incurred at the end-of-life. Bramley et al. [5] recently found that the health care costs of ABC patients in the US was about four times higher during the last six months before death than in the preceding months, with hospital admissions being one of the main driver for the increase. Even though the implications of high end-of-life costs–not only in ABC patients, but also in the general population–are subject to ongoing controversy [6,7], it must be noted that the phenomenon is poorly understood. Especially the combined variability of costs between and within patients over time is often not fully considered in the scientific literature [8].

ABC is a highly heterogeneous entity and patients differ considerably in their disease course and their lifetime costs [4]. Health care spending is also not constant, but fluctuates over time according to patients' needs and treatment plans. The analysis of 'average population trends' is therefore too simplistic to inform policy decisions regarding end-of-life care.

In this paper, we aim to explore the variability of longitudinal patterns of costs in ABC patients in the Netherlands during end-of-life care. First, we studied the average trend of costs in ABC patients over the last twelve months before death. We then identified latent groups of patients with distinct cost trajectories, solely based on the longitudinal data of their individual expenditures. Finally, we assessed whether the identified latent groups differed with respect to patient, tumor and treatment characteristics.

## Methods

### Patient and data collection

Our study is based on data of ABC patients from seven hospitals, which are a subset of the South East Netherlands Advanced Breast Cancer (SONABRE) Registry (NCT03577197) [9]. We included all individuals who were diagnosed with primary or recurrent ABC after January 2010 and died before June 1st 2017. Database lock was on October 23rd, 2017.

Information on patient and tumor characteristics, including hormone receptor (HR) and human epidermal growth factor receptor 2 (HER2) status were retrieved from the electronic hospital records by trained registration clerks. We also collected information on the units and the time of resource consumption at the participating hospitals. In particular, the following

resources were taken into account: medications (chemotherapeutic-, endocrine-, and targeted therapies, as well as bisphosphonates and the costs for in-hospital intravenous administration) and transfusions; local treatments and procedures (radiotherapy and surgical procedures, including among others, primary cancer, reconstructive and vascular access surgery); consultations and hospitalizations; and diagnostic procedures (imaging, biopsies, and testing of bio-marker CA 15–3 levels). The SONABRE Registry was approved by the Medical Research Ethics Committee of Maastricht University Medical Centre+. The need for informed consent was waived, because of the observational nature of this study.

## Allocation of costs over time

Whenever possible, we aimed to allocate costs to the days on which resources were consumed. Costs of systemic treatments were equally distributed between the first and the last day of administration. Because our study was based on routinely collected data, missing-ness was unavoidable. In some cases, the times of consumption were (partially) missing, and for certain resources, the exact dates of consumption were not registered and had to be inferred. This was mainly the case for diagnostic procedures, for which granularity of time information was limited. Respective costs could not be assigned to specific dates, but had to be attributed to consecutive treatment sequences. Moreover, in some instances, the quantities of resource consumption were unknown and had to be imputed [10]. A detailed description of the employed methods is provided in S1 File in the Electronic Supplementary Material.

Prior to the analysis, we aggregated costs to the monthly level. This allowed to better assess trends over time, as costs on the daily level were heavily affected by single events of individual patients. Costs were computed using Dutch guideline prices and expressed in €2017. If necessary, costs were inflated using the consumer price index [11]. When guideline prices were not available, prices were retrieved from other sources, including internal cost prices from the Maastricht University Medical Centre +, which are confidential and cannot be reported. All other unit prices are provided in S1 Table in the Electronic Supplementary Material. In addition to monthly costs, we also computed the total end-of-life costs per ABC patient, i.e. the costs incurred over the last twelve months before death, or, if survival time was shorter, between ABC diagnosis and death.

## Statistical analysis

Statistical analyses were conducted in four sequential steps. First, we descriptively assessed the average trend in costs for the entire cohort of deceased ABC patients over the last twelve months before death. Second, we used group-based trajectory modeling (GBTM) to identify latent groups of patients with distinct cost trajectories [12]. Third, we analyzed the composition of these latent groups by comparing both patient- and treatment-associated characteristics across them. Finally, we constructed a multiple, multinomial logistic regression model, in order to investigate the independent effects of the observed variables on patients' membership in latent classes. To account for the uncertainty of GBTM's class-assignment, the regression model linking cost trajectories' membership to covariates was weighted by subjects' posterior probabilities of assignment [13]. Because of the exploratory nature of this study, we did not adjust the significance level for multiple testing. P-values ≤0.05 were considered statistically significant.

**Longitudinal patterns in costs during the last twelve months before death.** We assessed the average trends in monthly hospital costs of ABC patients over the last twelve months before death. If the time between ABC diagnosis and death was shorter, patients only contributed

costs to the days and months in which they were diagnosed and alive (e.g. only to the last 15 days and 9-months). The costs of partially observed months (e.g. 15 days) following the diagnosis of ABC reflect the unadjusted sum of the daily costs. Results are presented for overall costs, as well as for costs of different resource categories.

**Identification of latent cost trajectories.**   We used GBTM to investigate latent cost trajectories in ABC patients over the last twelve months before death. GBTM is an unsupervised model-based clustering technique, designed to identify latent patterns of temporal change [12]. Groups are not specified ex ante (e.g. based on clinical characteristics), but are determined empirically from the cost data: individuals sharing a similar cost trajectory were grouped together. The basic rationale is to cluster individuals in groups that maximises both within-groups commonalities and between-groups differences. The longitudinal patterns of the resulting groups are referred to as latent trajectories (because they are not directly observable). We fitted the cost data of ABC patients using zero-inflated Poisson models, to account for excess zeros. To determine the optimal number of groups and the degrees of the polynomial function, we used a brute force approach: we fitted models with up to nine latent groups and up to quintic polynomials. For each model, we assessed the Akaike (AIC) and Bayesian Information criterion (BIC), as well as the absolute error in leave-one-out cross-validation (LOOCV), as proposed by Nielsen et al. [14]. After selecting the final model, posterior probabilities of belonging to each latent trajectory groups were computed and patients were assigned to the group, for which they had the highest posterior probability. Since the aggregation of costs across patients with different survival times could, potentially, create artificial longitudinal patterns, we conducted a sensitivity analysis in the subgroup of patients with at least 12 months survival time. Latent cost trajectory groups were extracted analogous to the methodology used in the full cohort model (for simplicity, LOOCV was not conducted and model selection was based on AIC and BIC alone). Results are provided in S42013S6 Figs and S3 and S4 Tables. A detailed description of the applied statistical methods is beyond the scope of this paper. For further information on GBTM and LOOCV we suggest Nagin et al. [12], Nielsen et al. [14], and the documentation of the 'crimCV' R-package [15] that was used for the analysis.

**Profile of latent cost trajectory groups.**   Identified latent groups were profiled by comparing the distribution of patient- and treatment-associated factors across them. The following characteristics were available and considered relevant: age, survival time (time between the date of ABC diagnosis and death), metastatic sites (any time), initial HR and HER2 receptor status, and chronic comorbidities (other malignancy, metabolic, cardio-vascular, or pulmonary disease, diagnosed at any time), as well as the las type of ABC treatment that was administered (chemo-, hormonal-, and/or targeted therapy; locoregional radical). To assess the statistical significance of the differences in categorical and continuous variables, we conducted Pearson's Chi-Square and Kruskall-Wallis tests, respectively.

**Multinomial logistic model.**   A multinomial logistic model was fitted to investigate whether patient- and treatment-associated factors explained patients' membership in latent cost trajectory groups independent from each other. Costs were not considered in the model, and the variable 'survival less than 12 months' was also excluded, because of high multicollinearity with the continuously measured survival time. All other independent variables were entered into the model. The analysis was explorative, and the model was not informed by our prior expectations. Instead, we used backward elimination to find the model with the lowest AIC. For each patient, we used the posterior-probabilities of group membership as weights in the model to account for GBTM's probabilistic nature of group assignment. P-values of coefficients were calculated using the Wald z-test. The analysis was conducted using the 'nnet' R-package [16].

# Results

## Sample characteristics

Data of 558 patients from the SONABRE Registry were analyzed. The mean end-of-life costs per patient were €21,641 (SD = 20,147). A total of 234 (42%) patients had a survival time of less than twelve months after the diagnosis of advanced disease. Further descriptive statistics are reported in Table 1.

**Table 1. Descriptive statistics.**

| Variable | Value |
|---|---|
| **Patient-associated factors** | |
| Age (years)—mean (SD) | 64 (14) |
| Survival time (days)a—median (95%CI) | 466 (421;533) |
| < 12 months survival time—n (%) | 234 (42) |
| Sites of metastasesb—mean (SD) | 2.1 (1.0) |
| Initial receptor status | |
| HR+/HER2-—n (%) | 355 (64) |
| HR+/HER2+—n (%) | 57 (10) |
| HR-/HER2+—n (%) | 45 (8) |
| TN—n (%) | 101 (18) |
| Comorbidities | |
| Metabolic disease—n (%) | 86 (15) |
| Cardio-vascular disease—n (%) | 64 (11) |
| Other malignancy—n (%) | 60 (11) |
| Pulmonary disease—n (%) | 42 (8) |
| **Treatment-associated factors** | |
| Locoregional radical treatmentc—n (%) | 39 (7) |
| Death in hospital—n (%) | 127 (23) |
| Intravenous last systemic treatmentd—n (%) | 218 (39) |
| Type of last treatment | |
| Chemotherapy only—n (%) | 200 (36) |
| Hormonal therapy only—n (%) | 157 (28) |
| Targeted-based therapye—n (%) | 113 (20) |
| No systemic therapy—n (%) | 88 (16) |
| **Costs** | |
| End-of-life costs (12 months)—mean (SD) € | 21,641 (20,147) |
| **Sample–n** | **558** |

[a] Survival time in days from the diagnosis of advanced disease;

[b] Sites of metastases indicates the number of different organ systems that are affected by metastases (e.g. brain, bone, visceral);

[c] locoregional radical treatment was defined as mamma surgery or radiotherapy with 15 or more fractions within the first year after diagnosis of advanced disease;

[d]intravenous last treatment = indicates whether the last treatment that was received was given intravenously;

[e] targeted-based therapy = targeted therapy with or without chemo or hormonal therapy; HR = Hormone receptor, HER2 = Human epidermal growth factor receptor 2; TN = triple negative (HR-/HER2-)

## Longitudinal patterns in costs during the last twelve months before death

Over the last twelve months before death, the average costs per patient month were €2,255 with a standard deviation of €492. The average longitudinal pattern of costs is shown in Fig 1. Two distinct phases can be distinguished: in the first phase, from month 12 until month 5 before death, the overall costs remained relatively stable (mean = €1,984). In the second phase, beginning in month 5, mean costs per month steadily increased with an average slope of €343 per month and reached a maximum of €3,614 per month during the last month before death. The transition between these phases were driven by two processes: a decrease in the costs for medications, starting from month 3, and an increase in costs for hospitalization and consultation, starting in month 5 before death. Hospitalizations became the main driver of costs after month 3, and costs reached a maximum during the last month before death (2,635). The contribution of diagnostic procedures and local treatments to total costs were marginal and did not show a clear trend over the last twelve months before death.

## Identification of latent cost trajectories

For the selection of the number of underlying latent classes, model-fit statistical criteria, BIC and AIC values, as well as the LOOCV prediction errors, did not unambiguously indicate the superiority of a particular model. After considering parsimony and assessing the relative changes in model fit at each step, we decided on a model with six latent groups and cubic polynomials. The specification of the final zero-inflated Poisson model is provided in S2 Table and detailed model evaluation results are shown in S1–S3 Figs in the Appendix.

The results of the final GBTM are shown in Fig 2. Six latent cost trajectory groups were identified, which we labeled according to their shapes: 1. MCI (moderate onset, continuously increasing), 2. HSD (high onset, slightly decreasing), 3. MFEM (moderate onset, fluctuating, early maximum), 4. MFLM (moderate onset, fluctuating, late maximum), 5.LSPE (low onset, stable with peak at the end), and 6. LSL (low onset, stable low). None of the identified latent

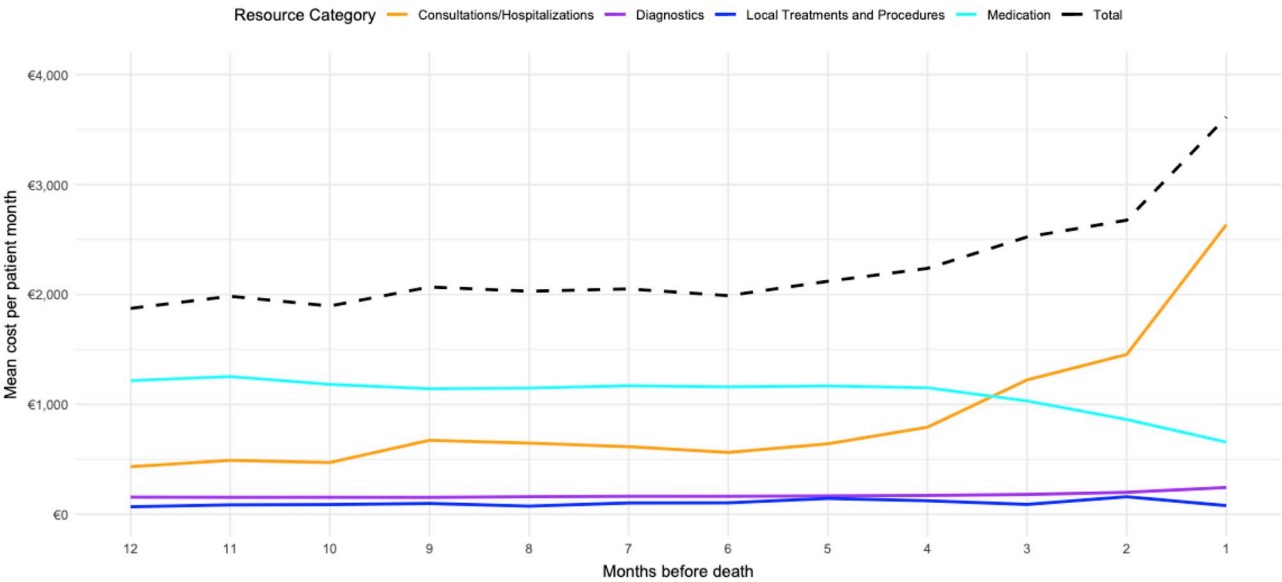

**Fig 1. Average longitudinal trend in monthly hospital costs of ABC patients over the last twelve months before death.** Shown are the total and resource-specific costs. The X-axis indicates the months until death.

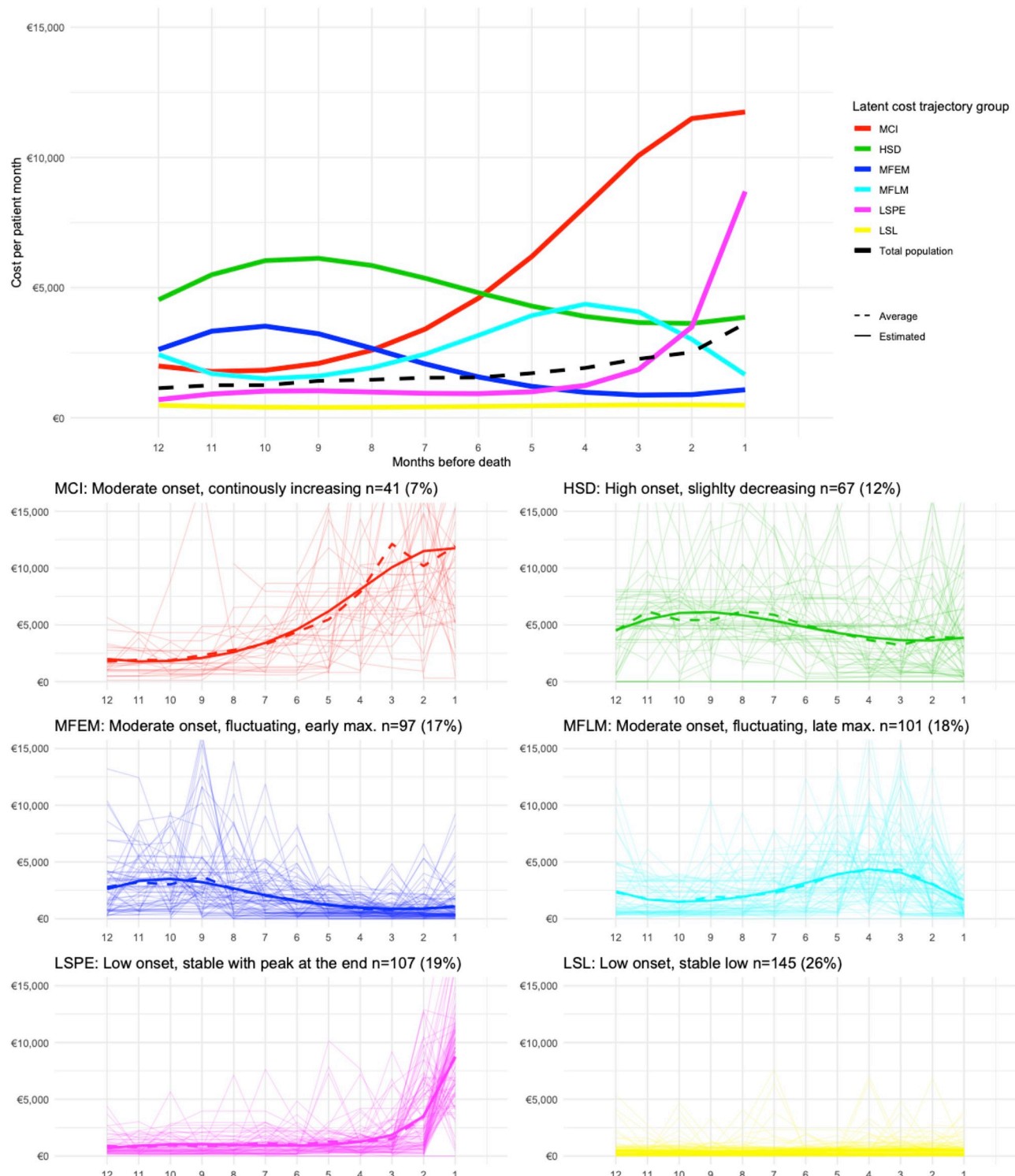

**Fig 2. Group-based trajectory modeling—Latent cost trajectory groups.** The top plot shows the results of the fitted GBTM model with six latent cost trajectory groups and cubic polynomials. For comparative purposes, the overall average trend (from Fig 1) is also shown. Below, the mean trajectories for each latent group (observed = dotted, estimated = solid line), in combination with the observed trajectories of the individual patients are presented.

group trajectories reflected the overall average trend, and only in groups MCI (n = 41) and LSPE (n = 107) we observed a marked increase in costs during the last months before death.

A sensitivity analysis in the subgroup of patients with at least twelve months survival time (n = 324) was conducted. Six latent cost trajectory groups were extracted, which were very similar to the groups found in the full cohort model. With a weighted Cohen's kappa of 0.79, the class allocation agreement was judged to be substantial. However, the peak in costs in the last month before death was less pronounced, and minor differences in latent trajectory patterns were observed. The detailed results are in S4–S6 Figs and S3 and S4 Tables. Overall, the results of the sensitivity analysis indicate that trajectory extraction was not materially affected by missingness.

## Profile of latent cost trajectory groups

Table 2 compares the distributions of observable patient- and treatment-associated factors across latent trajectory groups. The cells are color coded, depending on their relative values.

Group MCI was formed by a small group of patients (n = 41; 7%) and had the highest end-of-life costs, while their survival time was the shortest. Of these patients, 51% died within one year after diagnosis, 51% died in the hospital and 49%. received targeted therapy. Group HSD (n = 67; 12%) comprised of the youngest patients and had a high proportion of triple-negative ABC, and high use of chemotherapy and intravenously administered drugs as their last line of treatment. Groups MFEM (n = 97; 17%), MFLM (n = 101; 18%), and LSPE (n = 107; 19%) did not show prominent features. Group LSL stands out from the other groups in several aspects. This group was the largest (n = 145; 26%), and had the lowest end-of-life costs. The mean age in this group was the highest, and patients in this group had more frequently HR+/HER2-tumors, and the longest overall survival time. Members tended to have more pulmonary co-morbidities, less metastatic sites, more frequently endocrine therapy, or no systemic therapy at all, and a low rate of in-hospital deaths when compared with the other trajectories.

Overall, the identified latent groups did not only show considerable variation in their cost trajectories, but also differed significantly with respect to many patient- and treatment-related factors.

## Multinomial logistic model

The results of the multinomial model are shown in Table 3. From the twelve variables that were considered, seven were retained in the final model after backward elimination: age, number of metastatic sites, death in hospital, metabolic co-morbidity, intravenous last treatment and type of last treatment.

For some groups, the model indicated strong relationships between patient- and treatment-associated factors and group membership. The odds for being assigned to group MCI (the group with the highest end-of-life costs), instead of group LSL, were 6.7-times higher in patients who died in the hospital, and 8.8 times higher in patients who received a targeted therapy in the last months before death, for example. However, the overall fit of the model was poor. The McFadden pseudo $R^2$ was 0.12 and the model's predictive accuracy was limited: group membership was classified correctly in 38.5% (95% confidence interval = 34.5%–42.7%) of the patients, which is only marginally better than a 'null model', i.e. a model without predictors, which had an accuracy of 26.0%.

## Discussion

This study explored the longitudinal cost trajectories of ABC patients in seven hospitals in the Netherlands over the last twelve months before death. We found that, on average, costs

**Table 2. Distribution of clinical characteristics across latent cost trajectory groups—Mean (SD) or n (%).**

*Patient characteristics*

| | MCI | HSD | MFEM | MFLM | LSPE | LSL | p |
|---|---|---|---|---|---|---|---|
| Age | 61.8 (12.7) | 57.6 (16.2) | 65.4 (11.9) | 63.3 (13.0) | 63.0 (13.3) | 69.5 (15.9) | <0.001* |
| Survival time (days)a | 539 (543) | 602 (497) | 604 (472) | 525 (475) | 471 (473) | 688 (487) | <0.001* |
| < 12 months survival | 21 (51%) | 28 (42%) | 38 (39%) | 45 (45%) | 56 (52%) | 46 (32%) | 0.023* |
| Metastases | | | | | | | |
| Sites of metastasesb | 2.2 (1.0) | 2.1 (1.0) | 2.2 (0.9) | 2.3 (1.0) | 2.2 (1.1) | 1.8 (0.9) | <0.001* |
| Initial receptor status | | | | | | | 0.002* |
| HR+/HER2- | 23 (56%) | 35 (52%) | 61 (63%) | 58 (57%) | 70 (65%) | 108 (74%) | |
| HR+/HER2+ | 5 (12%) | 10 (15%) | 13 (13%) | 13 (13%) | 6 (6%) | 10 (7%) | |
| HR-/HER2+ | 6 (15%) | 4 (6%) | 9 (9%) | 16 (16%) | 4 (4%) | 6 (4%) | |
| TN | 7 (17%) | 18 (27%) | 14 (14%) | 14 (14%) | 27 (25%) | 21 (14%) | |
| Comorbidities | | | | | | | |
| Metabolic | 5 (12%) | 15 (22%) | 14 (14%) | 12 (12%) | 21 (20%) | 19 (13%) | 0.315 |
| Cardio-vascular | 4 (10%) | 5 (7%) | 15 (15%) | 7 (7%) | 12 (11%) | 21 (14%) | 0.301 |
| Other malignancy | 1 (2%) | 10 (15%) | 9 (9%) | 8 (8%) | 14 (13%) | 18 (12%) | 0.282 |
| Pulmonary | 2 (5%) | 4 (6%) | 4 (4%) | 10 (10%) | 5 (5%) | 17 (12%) | 0.156 |

*Treatment characteristics*

| | MCI | HSD | MFEM | MFLM | LSPE | LSL | p |
|---|---|---|---|---|---|---|---|
| Locoregional radicalc tx | 3 (7%) | 5 (7%) | 9 (9%) | 3 (3%) | 10 (9%) | 9 (6%) | 0.497 |
| Death in hospital | 21 (51%) | 12 (18%) | 14 (14%) | 18 (18%) | 44 (41%) | 18 (12%) | <0.001* |
| Intravenous last txd | 25 (61%) | 31 (46%) | 35 (36%) | 58 (57%) | 49 (46%) | 20 (14%) | <0.001* |
| Type of last tx | | | | | | | <0.001* |
| Chemo tx only | 11 (27%) | 34 (51%) | 39 (40%) | 35 (35%) | 46 (43%) | 35 (24%) | |
| Hormonal tx only | 5 (12%) | 8 (12%) | 26 (27%) | 17 (17%) | 28 (26%) | 73 (50%) | |
| Targeted-basede tx | 20 (49%) | 16 (24%) | 15 (15%) | 36 (36%) | 18 (17%) | 8 (6%) | |
| No systemic tx | 5 (12%) | 9 (13%) | 17 (18%) | 13 (13%) | 15 (14%) | 29 (20%) | |
| End-of-life costs (12 months) | 53,112 (22,816) | 44,352 (27,131) | 18,501 (10,897) | 24,098 (12,882) | 18,956 (8,263) | 4,619 (3,655) | <0.001* |
| Sample n | 41 | 67 | 97 | 101 | 107 | 145 | |

The table shows the distributions of patient- and treatment associated factors across latent trajectory groups. To allow for easier identification of differences, cells are color-coded, depending on the distance (in standard deviations) of the individual values from the row means. Red indicates positive, and yellow/white indicates negative deviations.

Tx = treatment;

a Survival time in days from the diagnosis of advanced disease;

b Sites of metastases indicates the number of different organ systems that are affected by metastases (e.g. brain, bone, visceral);

c locoregional radical treatment was defined as mamma surgery or radiotherapy with 15 or more fractions within the first year after diagnosis of advanced disease;

dIntravenous last treatment = indicates whether the last treatment that was received was given intravenously;

etargeted-based tx = targeted therapy with or without chemo or hormonal therapy; HR = Hormone receptor, HER2 = Human epidermal growth factor receptor 2; TN = triple negative (HR-/HER2-)

increased towards the end-of-life. In particular, we observed a marked rise in costs over the last five months, with a maximum in the last month preceding death. The increase was driven by inpatient admissions, while costs for medication decreased over the last three months of life. However, we identified six latent groups of patients with distinct longitudinal patterns of costs. All latent trajectories were strikingly different from the average trend, and, for a majority of the included ABC patients, costs did, in fact, not markedly increase towards the end-of-life. In a subgroup analysis, we found these results to be robust: similar latent cost trajectory patterns were uncovered in the subgroup of patients with at least twelve months of survival time.

**Table 3. Multinomial log-linear model.** Odds ratios and 95% confidence intervals for membership in cost trajectory groups–reference: LSL.

| | MCI | HSD | MFEM | MFLM | LSPE | LSL |
|---|---|---|---|---|---|---|
| Age (years) | 0.98 (0.95; 1.01) | 0.95* (0.93; 0.97) | 0.99 (0.97; 1.01) | 0.99 (0.96; 1.01) | 0.97* (0.95; 0.99) | Reference |
| Survival timea (years) | 0.69* (0.50; 0.95) | 0.69* (0.53; 0.88) | 0.76* (0.61; 0.94) | 0.61* (0.48; 0.77) | 0.57* (0.44; 0.72) | Reference |
| Number of metastatic sitesb | 1.33 (0.88; 2.01) | 1.21 (0.84; 1.72) | 1.45* (1.07; 1.98) | 1.60* (1.17; 2.21) | 1.39* (1.01; 1.89) | Reference |
| Death in hospital | 6.66* (2.82; 15.75) | 1.37 (0.58; 3.23) | 1.06 (0.49; 2.31) | 1.32 (0.61; 2.86) | 4.31* (2.19; 8.48) | Reference |
| Metabolic co-morbidity | 1.59 (0.50; 5.06) | 4.04* (1.71; 9.51) | 1.59 (0.72; 3.50) | 1.55 (0.66; 3.63) | 2.32* (1.07; 5.02) | Reference |
| Intravenous last treatmentd | 4.73*(1.48; 15.09) | 1.84 (0.77; 4.40) | 2.19 (0.95; 5.07) | 4.80* (1.97;11.70) | 4.10* (1.69; 9.93) | Reference |
| Last treatment | | | | | | |
| Chemotherapy only | Reference | Reference | Reference | Reference | Reference | Reference |
| Hormonal therapy only | 0.72 (0.17; 3.10) | 0.17* (0.06; 0.49) | 0.55 (0.24; 1.25) | 0.69 (0.26; 1.79) | 0.78 (0.32; 1.87) | Reference |
| Targeted- based therapye | 8.81*(2.81; 27.55) | 2.53 (0.91; 7.07) | 1.94 (0.71; 5.32) | 4.90*(1.86; 12.91) | 1.83 (0.65; 5.12) | Reference |
| No systemic therapy | 1.18 (0.25; 5.63) | 0.38 (0.13; 1.11) | 0.72 (0.27; 1.88) | 0.86 (0.29; 2.51) | 0.62 (0.22; 1.73) | Reference |
| Sample–n | 41 | 67 | 97 | 101 | 107 | 145 |

* = p≤0.05

[a] Survival time in days from the diagnosis of advanced disease;

[b] Sites of metastases indicates the number of different organ systems that are affected by metastases (e.g. brain, bone, visceral);

[d] intravenous last treatment = indicates whether the last treatment that was received was given intravenously; targeted-based tx = targeted therapy with or without chemo or hormonal therapy.

Descriptive results initially suggested that differences between the six cost trajectory groups might be governed by clinical aspects, but the goodness of fit of the final multinomial logistic model, including the patient- and treatment-associated factors, was relatively poor, and latent group membership remained largely unexplained.

While in this study, average costs of ABC patients increased as of five months before death, Bramley et al. [5] as well as Chastek et al. [17] found, in the context of the US health system, that costs only began to rise significantly about two months before death. In conformity with our findings, hospital inpatient care was the main driver of this increase in costs near the end-of-life. The general trend of an increase in health care costs towards the end of life is not limited to ABC patients, but also widely observed in other patient populations [5, 7].

GBTM has rarely been used to model health care costs [8,18,19], and, to the best of our knowledge, we were the first to explore the cost trajectories in ABC at the end-of-life with this technique. While previous studies focused exclusively on overall average trends or on costs in observable subgroups (e.g. based on age or cancer phenotype), in our study latent groups of patients were identified using the GBTM, solely based on their individual cost trajectories. It seems important to stress the difference in the conceptual approach: we did not intend to investigate to what extent the cost trajectories of, say, patients with HER2+ ABC differ from those of patients with HER2- ABC. Rather, we only looked at the cost trajectory data, in order to identify clusters of trajectories that are similar to each other. This allowed us to uncover hitherto hidden patterns of longitudinal costs in ABC patients. We found a considerable overlap of the cost trajectories of different ABC receptor subgroups: Patients with HER2+ and HER2-, for example, may share a very similar cost trajectory, and, even though it was a significant predictor in the bi- and multivariable analysis, the receptor status only explained a small proportion of the variance in the data.

Since we used data from patient hospital records, instead of administrative claims, which are commonly used to investigate health care costs [8,18], we were able to further investigate potential reasons for the uncovered variability in patients' cost trajectories. It should be noted that we did not aim to develop a model to predict a patient's cost trajectory in advance, say at

the date of diagnosis. The variables in the model, which in part could only be assessed after death, were not suitable for this purpose. Rather, we aimed to study to what extent patient- and treatment-related factors together could explain differences between patients' cost trajectory groups. Some of the associations found appear immediately intuitive: death in hospital, for example, independently increased the likelihood of patients to belong to a group with peak costs at the end of life (groups MCI and LSPE). However, the final multivariable model could also only partly explain latent group membership. This means, cost trajectories cannot easily be deduced from the clinical information used in this study.

This result is not necessarily surprising. Even though costs are largely driven by the choice of systemic therapy, which mainly depends on the cancer subtype, there is also room for individualized treatment decisions. Patients' and health care providers' preferences might therefore be an important cause for the substantial variation in cost trajectories. Moreover, courses of illness in ABC patients are complex and longitudinal costs might be predominantly determined by temporary dynamic factors, such as complications or disease progression [20]. Jiang et al. [19] drew on this point when they suggested that cost trajectories might even be considered as a parameter to monitor disease severity over time.

Our analysis is exploratory in nature, and findings should be interpreted within this context. An important point to stress is that our results cannot be generalized to a wider population of ABC patients. We included patients diagnosed since 2010 who died before June 2017, which makes our sample somewhat biased towards individuals with a shorter survival time when compared with the general ABC population. Furthermore, seven hospitals were included, which might also limit external validity. However, among the seven were academic, teaching and non-teaching hospitals, which should be considered a strength of this study. Also, our study had a strict hospital perspective, which implied that resources consumed in other areas of health care, such as general practitioners or hospices, were not included. Furthermore, the exact dates of consumption were not registered for all types of resources, which limited the precision of our cost trajectory estimates. However, since costs were aggregated to the monthly level, it is unlikely that more accurate time information would have led to different conclusions. It should be stressed that presented latent trajectory groups should not be understood as definite entities and that they cannot be observed directly, e.g. in clinical practice. It must also be noted that although GBTM results depend on statistical criteria, the number of latent groups and the degrees of polynomials depend–to some extent–on heuristics. Parsimony and the interpretability of the extracted trajectories also played a role in model building. This means that even though we think that the application of GBTM in similar settings (e.g. health insurance claims data, or ABC patients from other regions) will lead to similar conclusions, it is unlikely that the exact same latent trajectory groups will be uncovered. Finally, the used software package [15] did not allow to vary the polynomial orders between models. Specifically, for the Poisson count model, lower degree polynomials may provide a better fit.

Although immediate implications for practice are difficult to draw, there are lessons to be learned from our findings. The analyses showed that underlying the aggregate statistics, there is substantial variation in the cost trajectories of patients with ABC. The observed peak in the average costs before death was mainly attributable to a minority (26.5%) of patients (see Groups MCI and LSPE in Fig 2). This demonstrates that average population trend should not be mistaken as the cost trajectory of 'an average patient'–it cannot be used to make inferences about individual ABC patients, or even subgroups of patients [21].

A better understanding of the longitudinal dynamic structure of costs in ABC patients might help to improve the quality of economic evaluations and support better decision making in health policy. GBTM proved to be a promising and useful tool for exploring latent patterns in the cost data of ABC patients. To what degree the approach can lead to actionable insight

will be for future studies to determine. Those could build on our results and should aim to explain what causes the longitudinal patterns of costs. Even if differences in cost trajectories cannot (yet) be easily deduced from clinical characteristics, economic evaluations of treatments in ABC patients should aim to account for the heterogeneity of costs between and within patients over time.

## Conclusions

Average costs of ABC patients in the Netherlands increased over the last twelve months before death, mainly driven by hospital admissions. From this average trend, however, no inferences can be made with respect to individual patients or subgroups of patients. Our findings indicate that there are several latent groups of patients with distinct cost trajectories. None of them reflected the overall trend.

## Supporting information

**S1 Table. Unit costs.**
(DOCX)

**S2 Table. Final zero-inflated Poisson model—Beta coefficient point estimates.**
(DOCX)

**S3 Table. Final zero-inflated Poisson model—Beta coefficient point estimates for the subgroup of patients with at least 12 months survival time (n = 324).**
(DOCX)

**S4 Table. Latent group allocation agreement matrix.**
(DOCX)

**S1 File. Handling of missing data and vague time information.**
(DOCX)

**S1 Fig. The leave-one-out cross-validation (LOOCV) mean absolute error for fitted models with different numbers of latent groups (1–9) and different degrees of polynomials (1–5).** Lower values indicate better model fit. For nine groups, the model did not converge for polynomials greater 1.
(DOCX)

**S2 Fig. The Bayesian information criterion (BIC) values for fitted models with different numbers of latent groups (1–9) and different degrees of polynomials (1–5).** Lower values indicate better model fit. For nine groups, the model did not converge for polynomials greater 1.
(DOCX)

**S3 Fig. The Akaike information criterion (AIC) values for fitted models with different numbers of latent groups (1–9) and different degrees of polynomials (1–5).** Lower values indicate better model fit. For nine groups, the model did not converge for polynomials greater 1.
(DOCX)

**S4 Fig. Group-based trajectory modeling—Latent cost trajectory groups in the subgroup of patients with at least twelve months survival time (n = 324).** For convenience, each subgroup was colour-matched to the subgroup from the full cohort analysis, to which it was most similar. The top plot shows the results of the fitted GBTM model with six latent cost trajectory

groups and cubic polynomials. For comparative purposes, the overall average trend is also shown. Below, the mean trajectories for each latent group (observed = dotted, estimated = solid line), in combination with the observed trajectories of the individual patients are presented.
(DOCX)

**S5 Fig. The Akaike information criterion (AIC) values for fitted models with different numbers of latent groups (1–9) and different degrees of polynomials (1–5).** Results for the subgroup of patients with at least twelve months survival time. Lower values indicate better model fit. For six groups, the model did not converge for polynomials greater 3.
(DOCX)

**S6 Fig. The Bayesian information criterion (BIC) values for fitted models with different numbers of latent groups (1–9) and different degrees of polynomials (1–5).** Results for the subgroup of patients with at least twelve months survival time. Lower values indicate better model fit. For six groups, the model did not converge for polynomials greater 3.
(DOCX)

# Acknowledgments

We are sincerely grateful to the registration clerks of the SONABRE Registry for retrieving the data from patient medical records, to Tom Joorde from the Maastricht University Medical Centre+ for providing cost price information, and to Bram Wouterse from Netherlands Bureau for Economic Policy Analysis, for providing comments on an earlier version of this manuscript. We would also like to acknowledge all the patients whose data we have used for this study.

# Author Contributions

**Conceptualization:** Paul P. Schneider, Xavier G. L. V. Pouwels, Valéria Lima Passos, Bram L. T. Ramaekers, Sandra M. E. Geurts, Maaike de Boer, Frans Erdkamp, Birgit E. P. J. Vriens, Agnes J. van de Wouw, Marien O. den Boer, Manon J. Pepels, Vivianne C. G. Tjan-Heijnen, Manuela A. Joore.

**Data curation:** Paul P. Schneider, Xavier G. L. V. Pouwels, Sandra M. E. Geurts, Khava I. E. Ibragimova, Maaike de Boer, Frans Erdkamp, Birgit E. P. J. Vriens, Agnes J. van de Wouw, Marien O. den Boer, Manon J. Pepels, Vivianne C. G. Tjan-Heijnen.

**Formal analysis:** Paul P. Schneider, Xavier G. L. V. Pouwels, Valéria Lima Passos.

**Investigation:** Paul P. Schneider.

**Methodology:** Paul P. Schneider, Valéria Lima Passos, Bram L. T. Ramaekers, Sandra M. E. Geurts, Manuela A. Joore.

**Supervision:** Xavier G. L. V. Pouwels, Valéria Lima Passos, Bram L. T. Ramaekers, Vivianne C. G. Tjan-Heijnen, Manuela A. Joore.

**Validation:** Xavier G. L. V. Pouwels, Sandra M. E. Geurts, Khava I. E. Ibragimova.

**Visualization:** Paul P. Schneider.

**Writing – original draft:** Paul P. Schneider, Xavier G. L. V. Pouwels, Valéria Lima Passos, Bram L. T. Ramaekers, Manuela A. Joore.

**Writing – review & editing:** Paul P. Schneider, Xavier G. L. V. Pouwels, Valéria Lima Passos, Bram L. T. Ramaekers, Sandra M. E. Geurts, Khava I. E. Ibragimova, Maaike de Boer, Frans Erdkamp, Birgit E. P. J. Vriens, Agnes J. van de Wouw, Marien O. den Boer, Manon J. Pepels, Vivianne C. G. Tjan-Heijnen, Manuela A. Joore.

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
