## [Decision Letter · Decision Letter 0]

20 Dec 2019

PONE-D-19-22901

Variability of cost trajectories over the last year of life in patients with advanced breast cancer in the Netherlands

PLOS ONE

Dear Dr Schneider,

Thank you for submitting your manuscript to PLOS ONE. After careful consideration, we feel that it has merit but does not fully meet PLOS ONE’s publication criteria as it currently stands. Therefore, we invite you to submit a revised version of the manuscript that addresses the points raised during the review process.

We would appreciate receiving your revised manuscript by Feb 03 2020 11:59PM. To enhance the reproducibility of your results, we recommend that if applicable you deposit your laboratory protocols in protocols.io, where a protocol can be assigned its own identifier (DOI) such that it can be cited independently in the future. For instructions see: http://journals.plos.org/plosone/s/submission-guidelines#loc-laboratory-protocols

We look forward to receiving your revised manuscript.

Kind regards,

Academic Editor

PLOS ONE

Journal Requirements:

'The data collection of the Southeast Netherlands Advanced Breast Cancer Registry was funded by the Netherlands Organization for Health Research and Development (ZonMw), Novartis, Pfizer, Roche Nederland B.V., Eisai B.V., and Eli Lilly NL. The funders had no role in study design, data collection and analysis, decision to publish, or preparation of the manuscript.'

We note that you received funding from a commercial source: Novartis, Pfizer, Roche Nederland B.V., Eisai B.V., and Eli Lilly N.

Reviewers' comments:

Reviewer's Responses to Questions

**Comments to the Author**

1. Is the manuscript technically sound, and do the data support the conclusions?

Reviewer #1: Yes

Reviewer #2: Yes

Reviewer #3: Yes

2. Has the statistical analysis been performed appropriately and rigorously? 

Reviewer #1: Yes

Reviewer #2: Yes

Reviewer #3: Yes

3. Have the authors made all data underlying the findings in their manuscript fully available?

Reviewer #1: Yes

Reviewer #2: No

Reviewer #3: No

4. Is the manuscript presented in an intelligible fashion and written in standard English?

Reviewer #1: Yes

Reviewer #2: Yes

Reviewer #3: Yes

5. Review Comments to the Author

Reviewer #1: This research is a novel approach to the costing of end of life costs. This is an important research reminding that there are many different and so far unexplained cost-trajectories which are not considered by conventional health state-based cost-effectiveness analyses. The paper was written with scientific rigor as well as with an enjoyable flow. There is a growing body of literature regarding the determinants of end-of-life costs and the readers could potentially benefit from some additional discussion of the literature although I do not deem this as a necessity. This research is an important contribution to the analysis of healthcare costs.

Reviewer #2: Strength of this article

The significance of this article is that it goes beyond the average trajectory and explores subgroups with distinct trajectory patterns. Such patterns are often lost in an analysis that focuses on average trajectory.

The group-based trajectory modeling has been used in the literature and it is a useful method to explore the different trajectory patterns.

Weakness of this article

The sample size (n = 558) is a little bit small for this type of studies. Large datasets can often be obtained from, for example, insurance claims database, and breast cancer is not a rare disease among cancer patient population.

There is a little bit lack of depth in this article, because the possible reasons for the different trajectory shapes are not studied (I recognize that this is left to future work, but it does diminish the depth of this paper to some degree).

Even without looking at the conclusion of this article, one can speculate that each patient's cost trajectory is more or less different because of various patient-level, physician-level and hospital-level factors. Therefore, heterogeneity in cost trajectory exists. The number six, i.e., six pattern groups, is perhaps more or less subjective. If one has a much larger dataset, perhaps more patterns can be identified by the statistical criteria used in this paper. Without looking into the inside mechanism that leads to the different patterns, the conclusion and results of this article is superficial.

The conclusion says: "From this average trend, however, no inferences

can be made with respect to individual patients or subgroups of patients." I am skeptical about this statement. This paper identified six trajectory shapes. For an individual, certainly we can plot the cost trajectory of that person and determine which shape group this person's trajectory most likely belongs to.

The subsection "Longitudinal Patterns in Costs During the Last Twelve Months Before Death" says: "If the time between ABC diagnosis and death was shorter, patients only

contributed costs to the months in which they were diagnosed and alive (i.e. only to the last x months)." I disagree with this way of handling patients with less than 12 month of follow-up. It is better to focus only on patients with at least 12 months of follow-up. It is difficult to justify averaging over patients with different lengths of follow-up.

Clarification needed

The costs are aggregated by month. If a patient dies in the middle of the month, then the last month is not a full month, which drives down the cost. Please explain how this issue was handled.

Please explain how the time-stamp of the costs is determined. If a procedure is performed on a patient, did you attribute the cost to the date on which the procedure is performed, or to the date when the billing for that procedure is processed? If medication is prescribed, did you attribute the medication cost to the date of prescription or distribute the medication cost across all the days in the prescription period?

Page 14 of the paper says "We fitted the cost data of ABC patients using zero-inflated Poisson models, to account for excess zeros". Further details need to be provided about this model (maybe in appendix). First, how do you justify this is zero-inflated Poisson instead of the usual Poisson model? Second, how did you deal with overdispersion, a common difficulty with Poisson model? Third, this could be longitudinal zero-inflated Poisson, and accommodating the longitudinal nature of data (i.e., intra-subject correlation among the 12 monthly costs per patient) is not a computationally trivial task. How did you handle this situation in the software that you use? It would be better to present the most important model information without asking the reader to read the documentation of crimCV package.

Page 14. Please clarify the time of measurement for some covariates: age, co-morbidity, and type of ABS treatment (what if the treatment changes over time?). Please also clarify how the survival time is calculated if every patient has 12 month follow-up.

Reviewer #3: Schneider et al. analyzed health care utilization data of 558 advanced breast cancer cases in the Netherlands using group-based trajectory modelling. The authors noted interesting patterns. While it is generally assumed that end-of-life health care expenditures follow a hockeystick-like trajectory, their analysis revealed six distinct patterns with very differing shapes. Of note, only two out of the six patterns showed an increase towards the final life months. The patterns were largely driven by medication costs (until around month 6) and later by hospitalization (possibly indicating a switch of care focus to palliative care).

The analysis is generally sound and the paper is well written. I also like the approach the authors have undertaken: away from mean-based comparisons toward more comprehensive assessments of longitudinal costs. That said, I have three major comments for the author's consideration.

1) The cost data are not actual costs from health insurance claims but derived from health care use (specific medications, procedures), with acutal costs derived and aggregated from official sources. This has the advantage that one does not need to worry about inflation when analyzing multiple years. But this has also two downsides: costs accrued outside the hospital setting (e.g. palliative care) are not included (which is acknowledged in the discussion), but also a lot of the variability observed in claims data is removed. This latter point is particularly important here. It leads me to believe that the findings will likely be not reproducible when applying the same methodology to health care expenditure data. In fact, their finding of six distinct and quite robust groups may even be overly optimistic. Real-world (claims) data is messy and noisy. Therefore I would be surprised if the method performed equally well in other databases of similar size.

2) I am a little concerned about censoring due to death in the analysis. More than 40% of patients died in less than 12 months after diagnosis, which could have influenced trajectory shapes (e.g. the HSD and MFEM groups, which have high costs in the first three months after diagnosis). In other words, the observed trajectory patterns could somehow also originate from data availability. Does the statistical method offer possibilities to include censoring? If not, are there other means to verify that the observed groups are not artifacts of follow-up duration or missing data?

3) Advanced breast cancer is a condition where treatment strategies are dependent of genetic markers (HER2, Hormone receptors). It seems therefore a bit counterintuitive to group patients by cost trajectories rather than by genetic subgroups (which is what most physicians would do). Especially since health care utilization and expenditures differ by genetic marker / treatment group. Although this analysis is a nice illustration of cost trajectory analysis, it is probably not the typical use case for this type of method. I would probably resort to trajectory analysis if no other established subgroup definitions are available. On the other hand, the observation that certain genetic marker combinations are indeed predominant in some trajectories could be considered some kind of a validation. Can the authors elaborate for which situations /databases / settings they consider their trajectory analysis method particularly suitable?

6. PLOS authors have the option to publish the peer review history of their article (what does this mean?). If published, this will include your full peer review and any attached files.

Reviewer #1: Yes: NIKOLAOS KOTSOPOULOS

Reviewer #2: No

Reviewer #3: No

---

## [Author Response · Author response to Decision Letter 0]

7 Mar 2020

A Response to Reviewer Comments file has been uploaded separately.

---

## [Editor Report · Decision Letter 1]

12 Mar 2020

Variability of cost trajectories over the last year of life in patients with advanced breast cancer in the Netherlands

PONE-D-19-22901R1

Dear Dr. Schneider,

We are pleased to inform you that your manuscript has been judged scientifically suitable for publication and will be formally accepted for publication once it complies with all outstanding technical requirements.

With kind regards,

Academic Editor

PLOS ONE
---

## [Editor Report · Acceptance letter]

13 Mar 2020

PONE-D-19-22901R1 

Variability of cost trajectories over the last year of life in patients with advanced breast cancer in the Netherlands 

Dear Dr. Schneider:

I am pleased to inform you that your manuscript has been deemed suitable for publication in PLOS ONE. Congratulations! Your manuscript is now with our production department. 

With kind regards,

on behalf of

Dr. Hakan Buyukhatipoglu 

Academic Editor

PLOS ONE